# Analysis of SYK Gene as a Prognostic Biomarker and Suggested Potential Bioactive Phytochemicals as an Alternative Therapeutic Option for Colorectal Cancer: An In-Silico Pharmaco-Informatics Investigation

**DOI:** 10.3390/jpm11090888

**Published:** 2021-09-06

**Authors:** Partha Biswas, Dipta Dey, Atikur Rahman, Md. Aminul Islam, Tasmina Ferdous Susmi, Md. Abu Kaium, Md. Nazmul Hasan, MD. Hasanur Rahman, Shafi Mahmud, Md. Abu Saleh, Priyanka Paul, Md Rezanur Rahman, Md. Al Saber, Hangyeul Song, Md. Ataur Rahman, Bonglee Kim

**Affiliations:** 1Department of Genetic Engineering and Biotechnology, Faculty of Biological Science and Technology, Jashore University of Science and Technology (JUST), Jashore 7408, Bangladesh; partha_160626@just.edu.bd (P.B.); atikur.just@gmail.com (A.R.); aminul_180603@just.edu.bd (M.A.I.); tsusmi7@gmail.com (T.F.S.); 150623.gebt@student.just.edu.bd (M.A.K.); 2ABEx Bio-Research Center, East Azampur, Dhaka 1230, Bangladesh; hasanurrahman.bge@gmail.com; 3Laboratory of Pharmaceutical Biotechnology and Bioinformatics, Department of Genetic Engineering and Biotechnology, Faculty of Biological Science and Technology, Jashore University of Science and Technology (JUST), Jashore 7408, Bangladesh; mn.hasan@just.edu.bd; 4Department of Biochemistry and Molecular Biology, Life Science Faculty, Bangabandhu Sheikh Mujibur Rahman Science and Technology University, Gopalgonj 8100, Bangladesh; diptadey727@gmail.com (D.D.); paul.bmb011@gmail.com (P.P.); 5Fermentation Engineering, School of Biotechnology, Jiangnan University, Wuxi 214122, China; 6Department of Biotechnology and Genetic Engineering, Bangabandhu Sheikh Mujibur Rahman Science and Technology University, Gopalganj 8100, Bangladesh; 7Department of Genetic Engineering and Biotechnology, University of Rajshahi, Rajshahi 6205, Bangladesh; shafimahmudfz@gmail.com (S.M.); saleh@ru.ac.bd (M.A.S.); 8Queensland Brain Institute, The University of Queensland, Brisbane, QLD 4072, Australia; rezanur.rahman@uq.net.au; 9Biotechnology, University of Pécs, Medical School, 7624 Pécs, Hungary; TMFDPP@pte.hu; 10Department of Pathology, College of Korean Medicine, Kyung Hee University, 26 Kyungheedae-ro, Dongdaemun-gu, Seoul 02447, Korea; shg3811@khu.ac.kr; 11Korean Medicine-Based Drug Repositioning Cancer Research Center, College of Korean Medicine, Kyung Hee University, Seoul 02447, Korea; 12Global Biotechnology & Biomedical Research Network (GBBRN), Department of Biotechnology and Genetic Engineering, Faculty of Biological Sciences, Islamic University, Kushtia 7003, Bangladesh

**Keywords:** SYK gene, onco-informatics, colorectal cancer, biomarker, molecular dynamics simulation, colorectal adenocarcinoma

## Abstract

Background: SYK gene regulates the expression of SYK kinase (Spleen tyrosine kinase), an important non-receptor protein-tyrosine kinase for immunological receptor-mediated signaling, which is also considered a tumor growth metastasis initiator. An onco-informatics analysis was adopted to evaluate the expression and prognostic value of the SYK gene in colorectal cancer (CRC), the third most fatal cancer type; of late, it may be a biomarker as another targeted site for CRC. In addition, identify the potential phytochemicals that may inhibit the overexpression of the SYK kinase protein and minimize the human CRC. Materials & Methods: The differential expression of the SYK gene was analyzed using several transcriptomic databases, including Oncomine, UALCAN, GENT2, and GEPIA2. The server cBioPortal was used to analyze the mutations and copy number alterations, whereas GENT2, Gene Expression Profiling Interactive Analysis (GEPIA), Onco-Lnc, and PrognoScan were used to examine the survival rate. The protein-protein interaction network of SYK kinase and its co-expressed genes was conducted via Gene-MANIA. Considering the SYK kinase may be the targeted site, the selected phytochemicals were assessed by molecular docking using PyRx 0.8 packages. Molecular interactions were also observed by following the Ligplot+ version 2.2. YASARA molecular dynamics simulator was applied for the post-validation of the selected phytochemicals. Results: Our result reveals an increased level of mRNA expression of the SYK gene in colorectal adenocarcinoma (COAD) samples compared to those in normal tissues. A significant methylation level and various genetic alterations recurrence of the SYK gene were analyzed where the fluctuation of the SYK alteration frequency was detected across different CRC studies. As a result, a lower level of SYK expression was related to higher chances of survival. This was evidenced by multiple bioinformatics platforms and web resources, which demonstrated that the SYK gene can be a potential biomarker for CRC. In this study, aromatic phytochemicals, such as kaempferol and glabridin that target the macromolecule (SYK kinase), showed higher stability than the controls, and we have estimated that these bioactive potential phytochemicals might be a useful option for CRC patients after the clinical trial. Conclusions: Our onco-informatics investigation suggests that the SYK gene can be a potential prognostic biomarker of CRC. On the contrary, SYK kinase would be a major target, and all selected compounds were validated against the protein using in-silico drug design approaches. Here, more in vitro and in vivo analysis is required for targeting SYK protein in CRC.

## 1. Introduction

SYK gene encodes Spleen tyrosine kinase (SYK) non-receptor protein-tyrosine kinase, which is mainly associated with adaptive immune receptor signaling and involved in various signaling-mediated biological processes, including innate immunity recognition, platelet activation, and vascular development, along with cellular responses, such as differentiation, proliferation, and phagocytosis [1]. In a human cell, the SYK gene is located at the q22.2 position of the 9th chromosome. The SYK gene is broadly communicated in hematopoietic cells, and it is a capable modulator of epithelial cell development and a potential tumor target in human cancers. The expression of the SYK gene is equally important for both normal and cancer cells. In contrast, it has a dual nature processing both tumor-promoting as well as tumor-suppressing capabilities. Various shreds of evidence suggest that the SYK gene is involved in cancer formation [2]. Tumor growth and metastasis are developed and activated through the SYK signaling pathway as an intermediate [3]. The SYK gene is considered a potent tumor suppressor gene in humans, and its activity increases the suppression of tumorigenicity of breast cancer (BC) [2,3,4,5,6]. A lower SYK protein expression and methylation are also connected with the metastasis of other cancer, including lung cancer, liver cancer, oral squamous cell cancer, pancreatic cancer, bladder cancer, gastric cancer, and urinary cancer [7,8,9]. Here, colorectal cancer (CRC) is also associated with this gene, and the loss of its expression is found in several CRC tissues [10,11].

Colorectal cancer (CRC) is the third most commonly diagnosed and the second most common cause of cancer-related deaths worldwide in both sexes and across all ages. According to the GLOBOCAN 2020 data, around 2 million people (10%) were diagnosed, and approximately 1 million people (9.4%) died from CRC in both sexes and all ages worldwide [12]. CRC is a global concern, and its burden is expected to escalate by about 60% in countries with high or very high human development index, particularly in Eastern Europe, Asia, and South America by 2030 [13]. Thus, it is necessary to establish improved assessment and treatment strategies; however, its therapy and prognosis widely vary with various malignancy stages and biological features. Based on the tumor node metastasis staging, stage I CRC patients have a five-year endurance pace of >90%, while it declines by 12% [14]. CRC is a heterogeneous group of diseases caused by genetic and epigenetic mutations in certain epithelial cells [15]. Because of the complexity of the CRC disease, a more comprehensive method is required to detect the CRC accurately with high accuracy with rapidly based on a bioinformatics-based test. In this regard, a multi-omics data mining approach can be a feasible alternative to identify the target and, specifically, anti-cancer therapy approaches. The SYK gene possesses dual characteristics of a tumor promoter gene within various lymphomas, leukemias, carcinomas, and cancer types (lung cancer, ovarian cancer, and small cell lung cancer). In contrast, it acts as a tumor suppressor gene within several malignant tumors, namely-melanoma, gastric cancer, bladder cancer, hepatocellular carcinoma, and many more [5]. Through the signaling of the B-cell receptor, SYK acts as a key modulator in various B cell-driven lymphomas. Contrarily, in B-cell lymphocytic leukemia (B-CLL), diffuse large B-cell lymphoma (DLBCL), follicular lymphoma, mantle cell lymphoma, marginal zone lymphoma (MZL), B-cell acute lymphocytic leukemia (B-ALL), acute myeloid leukemia (AML), SYK performed as a tumor promoter gene [5]. The latest cancer therapies have severe side effects, high costs, and single survival thresholds [16]. In addition, with the development of drug resistance in cancer cells, the development of new therapeutic agents against cancer is a crying need in recent times. Several studies have shown that purified plant extracts or phytochemicals have significant positive results in cancer treatment when used directly or in combination with existing drugs [17]. Various well-known phytochemicals are found to exhibit a specific action against tumors, BC, and prostate cancer (PC) [18,19]. It is also shown that various plant-derived compounds have anti-proliferative activity against cancer by the modulation of cellular pathways that make the phytochemicals worthy as perfect drug candidates [20]. Plant bioactive compounds suppress the interaction of carcinogenic molecules that damage the DNA and are usually considered safe and easily available to consumers, so they are suggested as an alternative and effective therapeutics for cancer treatment with few side effects in the human body. Because of chemotherapy’s serious side effects and expensiveness, we have selected some aromatic phytochemicals, such as glabridin, curcumin, kaempferol, quercetin, genistein, against the Control drug capecitabine (CAP), which are safer within the human body and plays a pivotal role in diverse cancer types. Glabridin is a member of hydroxyisoflavans which have antitumor and antimetastatic effects on breast, liver, and lung cancers [21]. Curcumin is found to possess an anticancer activity and antitumor cell invasion by suppressing several signaling pathways in cancer [22] and also the compound Kaempferol exhibits its anticancer activity by preventing cell invasion, downregulating AKT phosphorylation, inhibiting the MMP-2 protein, and regulating the expression of other cancer-inducing pathways [23,24]. Quercetin also exerts pro-apoptotic effects on tumor cells, preventing the progression and suppressing the tumor growth of several cancer cell lines, such as breast, lung, ovarian, colorectal, and stomach cancer and leukemia [25]. Genistein, a soy-derived isoflavone, also has anticancer activity and significant regulatory effects on cell growth, including DNA mutation inhibition, cellular apoptosis, and tumor gene suppression [26]. CAP belongs to the class of antineoplastic agents named antimetabolites; it has anticancer activity and acts by slowing the growth of tumor cells by inhibiting the DNA synthesis [27,28]. Considering all the safety effects of the selected phytochemicals within the human body, bioinformatics-based analysis can represent better results against the SYK kinase under the molecular docking and molecular dynamics platform. It may minimize the severity of colorectal cancer.

In this study, an in-depth analysis was performed to examine the role and clinical relevance of the SYK gene in CRC development by adopting various cancer databases. Here, the gene expression patterns were determined in different malignancy types, clinicopathological parameters, methylation status, genetic alteration frequencies, survival assay analysis, genes co-expressed with SYK, and the interaction network of these genes. Moreover, preliminary aromatic phytochemicals were selected based on their pharmacokinetic properties. Consequently, further analysis (e.g., Molecular Docking, post-Docking interaction, and Molecular Dynamics Simulation-MDS) of these compounds with the soluble SYK kinase protein using in silico drug design approaches. These multi-omics data mining and in silico drug designing approaches will help us find new anticancer agents.

## 2. Methods and Materials

### 2.1. The Analysis of SYK Gene Expression in Colorectal Cancer

This study aimed to examine the mRNA transcription levels of various cancers, including CRC, using Oncomine (https://www.oncomine.org/, accessed on 2 February 2021), UALCAN (http://ualcan.path.uab.edu/, accessed on 2 February 2021), GENT2 (http://gent2.appex.kr/gent2/, accessed on 2 February 2021), and GEPIA2 (http://gepia2.cancer-pku.cn/#index, accessed on 2 February 2021) databases. These are all publicly accessible interactive online platforms with the mRNA transcriptional levels of various cancer in different malignancy tests compared to the control. Currently, Oncomine is the world’s biggest oncogene chip information database, and it incorporated the information on mining stages with 715 autonomous datasets and 86,733 examples [29]. The mRNA expression of the SYK gene in clinical diseases was contrasted and related to ordinary controls where the limit boundaries of the *p*-value and fold change were settled as 0.0001 and 2, respectively. An in-depth analysis of the SYK gene in colon adenocarcinoma (COAD) using the TCGA database was done from the UALCAN web portal, which provides the expression, methylation, pan-cancer view, survival, and correlation data [30]. The DNA methylation levels at the promoter region of the SYK gene based on various parameters were also picked out from the UALCAN database. The Gene Expression Database of Normal and Tumor tissues 2 (GENT2) was also employed to retrieve the data on SYK gene expression across 72 different paired tissues [31]. Moreover, the mRNA expression of the SYK gene and its expression in cancer stages using the TCGA data in COAD was also further assessed from the Gene Expression Profiling Interactive Analysis (GEPIA) 2 database [32]. It provides a gene-specific relative investigation of malignant growth types using a standard handling pipeline with around 8587 typical and 9736 tumor samples.

### 2.2. Mutation and Copy Number Alteration Determination in the SYK Gene

The cBioPortal for Cancer Genomics (https://www.cbioportal.org/, accessed on 2 February 2021) is an open-source interactive web portal to systematically explore multidisciplinary cancer genomic datasets currently containing 308 cancer studies [33,34]. It provides information from over 5000 tumor samples to map the recurrence of mutations and other hereditary modifications, atomic profiling of cell lines, and cancer tissues by applying multifaceted cancer studies. The cBioPortal was scrutinized to analyze the conversion and copy number alterations (CNA) in the SYK gene in CRC.

### 2.3. Survival Assay Analysis

The survival plot analysis is a statistical view that provides the data on the survival rate of cancer patients against time. Various web-based tools were employed, including GENT2, GEPIA, OncoLnc, and UALCAN, to examine the survival rate of CRC patients against the expression of the SYK gene. Here, the GENT2 website (http://gent2.appex.kr, accessed on 2 February 2021) was used to identify the survival condition where the analysis was performed based on cancer subtype and the prognosis condition for a specific tissue [31]. This database provides data for colon cancer based on 1146 samples by analyzing five subtypes, including the molecular subtype, AJCC stage, Duke stage, grade, and histology. GEPIA (http://gepia.cancer-pku.cn/, accessed on 2 February 2021) is a database that also explores the survival data for overall survival (OS) or disease-free survival (DFS) [35]. It uses a log-rank test and performs based on gene expression by selecting any specific cancer type [36]. The OncoLnc database (http://www.oncolnc.org/, accessed on 2 February 2021) was also used for survival analysis. It provides the survival data of 8647 patients by analyzing the mRNA, miRNA, or lncRNA expression of 21 cancers 18 [37]. Besides, the UALCAN database (http://ualcan.path.uab.edu, accessed on 2 February 2021) is an online resource used for the survival analysis of the SYK gene. It offers the relationship between gene expression and patient survival, evaluates the TCGA patient survival data for Kaplan–Meier survival analyses, and generates OS plots [30]. 

### 2.4. Correlation Analysis and Interaction Network

It is essential to classify the associated genes of the target gene to analyze the gene expression better. This was accomplished by looking at the related genes of the SYK gene on the UALCAN and GEPIA websites. The GEPIA server was used to detect similar genes for TCGA COAD tumors. Also, using the correlation segment of the UALCAN database, the genes positively correlated with SYK in COAD were identified, where Pearson correlation coefficients (CC) >0.47 was considered as moderately significant on both websites and recognized the similar or correlated genes were inputted into the GeneMANIA database to predict the protein-protein interaction network. GeneMANIA (https://genemania.org/, accessed on 2 February 2021) is an online database that indicates the interaction network of a set of genes with the inputted genes. The database provides the associated protein and genetic interactions, pathways, co-expression, co-localization, and protein domain similarity [31]. GeneMania was also used to predict the protein-protein interaction network of the SYK gene to express its relationship with the listed genes.

### 2.5. Target Identification of the SYK Gene for Colorectal Cancer

SYK is a non-receptor protein-tyrosine kinase that has a part of performing when combined with different malignancies; thus, this kinase has become an important target for developing a therapeutic drug. It has been reported that the loss or decreased SYK is expressed in different normal tissues and cancer cell types [38,39,40,41]. The failure of the SYK expression is the main cause of CRC development. Its expression could be restored in the CRC cell line using several selected drugs or might be phytochemicals. As cell proliferation is inhibited by the lower or overexpression of the SYK gene, it could be used as a therapeutic target against CRC by targeting the protein identified.

### 2.6. Pharmacokinetics Analysis

All selected aromatic compounds were initially analyzed via the “Swiss ADME (http://www.swissadme.ch/index.php, accessed on 2 February 2021) online server, which evaluated the physiochemical properties of the lead molecules, and every compound followed the Lipinski’s rule of five. The pharmacokinetics (absorption, distribution, metabolism, excretion, and toxicity) of the extracted ligand was analyzed using pKCSM as an online server (http://biosig.unimelb.edu.au/pkcsm/prediction, accessed on 2 February 2021)” [42].

### 2.7. Extracting Lead Molecule for Optimization

The phytochemicals were first retrieved from the PubChem database to prepare the ligand for the docking of protein-ligands, and the 3D phytochemical structure required for molecular docking was obtained. The compounds were prepared as ligands using the “UCSF Chimera 1.14 (https://www.cgl.ucsf.edu/chimera/, accessed on 2 February 2021) tool”, where the ligand was optimized, added with hydrogen atoms, and charged at zero; the output file was saved in Protein Data Bank (PDB) format [43]. 

### 2.8. Extraction of Proteins and Preparation for Docking

The tertiary, non-mutated structure of the soluble SYK kinase protein, which has the PDB ID (4XG3), was collected from the online PDB database (https://www.rcsb.org/, accessed on 2 February 2021), and the sequential steps for protein preparation were followed for molecular docking. Unnecessary things were eliminated, such as the water molecules, metal ions, ligands, heteroatoms, and unwanted side chains. Indispensable hydrogen bonds were also added to the macromolecules for more precise docking, and the total operation was conducted by the “UCSF Chimera package tool (Version-1.14)”, here must followed the “Gasteiger’s method” [44]. The prepared protein structures were saved in PDB format and conserved for further analysis [45].

### 2.9. Molecular Docking and Post-Docking Data Visualization

For optimizing a selected ligand, which was proposed using a drug target against the selected SYK receptor, protein-ligand docking was done using the tool “PyRx software package version 0.8” (https://pyrx.sourceforge.io/home, accessed on 2 February 2021) that performs based on the Auto-Dock Vina Platform. In each docking, the top 10 best binding affinities were displayed via (.csv) file [46,47]. 

Ligplot+ version 2.2 was used to analyze the interaction between protein-ligand complexes, where mainly the non-bond and non-covalent (polar and hydrophobic) interactions were evaluated. The Ligplot+ visualizing tool was effectively performed based on the java interface (Java SE Runtime Environment 8u271), where only the combined PDB files run retrieved from the PYMOL visualizing tool [48]. 

### 2.10. Molecular Dynamics Simulation

The molecular dynamics complex simulation was carried out in YASARA dynamics, where an AMBER14 force field [49] was used [50]. The cubic simulation cell was created, the complexes were optimized, and the hydrogen bond networks were oriented [51]. A TIP3P water solvation model was used (0.997 g/L^−1^, 25 c, 1 atm), where the steepest gradient algorithms were used by a simulated annealing method to minimize the protein complexes. The simulation system was neutralized at 0.9% NaCl, 310 K, and pH 7.4. The techniques of Particle Mesh Ewald were used to calculate the electrostatic interaction, with a cut-off radius of 8 Å [52]. On either side of the system, the simulation cell was expanded to 20 Å so that the protein could move freely inside [53]. To maintain the simulation temperature, a Berendsen thermostat was used. The simulation was conducted at 1.25 fs, and the trajectories were preserved every 100 ps [54]. The simulation was performed for over 100 ns, and trajectory analysis was carried out to measure the root-mean-square fluctuations (RMSF), root-mean-square deviations (RMSD), hydrogen bonds, the radius of gyration (Rg), and solvent-accessible surface areas (SASA).

## 3. Results

### 3.1. Expression Analysis of the SYK Gene

First, the differential expression patterns of the SYK gene in various cancer types were analyzed using the Oncomine, GENT2, and GEPIA2 databases. Significantly differential expression patterns of SYK mRNAs were found in multiple cancers compared to the control (Figure 1a). Notably, a significant upregulation of the SYK gene was found in CRC (*p* < 0.05) compared to the control. In total, there were 12 databases for CRC on Oncomine, and the statistical details of the SYK gene expression in different CRC subtypes from the Oncomine database are shown in Appendix A. According to the GEPIA2 database, the SYK gene was significantly upregulated in COAD compared to the paired normal tissues (Figure 1c).

Moreover, the SYK gene was also significantly upregulated in various cancer types, such as invasive breast carcinoma (BRCA), cervical squamous carcinoma and endocervical adenocarcinoma (CESC), cholangiocarcinoma (CHOL), squamous cell carcinomas of the head and neck (GBM), acute myeloid leukemia (LAML), rectum adenocarcinoma (READ), stomach adenocarcinoma (STAD), and uterine corpus endometrial carcinoma (UCEC) (Figure 1b). On the contrary, the SYK gene was downregulated in bladder urothelial carcinoma (BLCA), kidney renal clear cell carcinoma (KIRC), kidney renal papillary cell carcinoma (KIRP), prostate adenocarcinoma (PRAD), cutaneous skin melanoma (SKCM), and thyroid carcinoma (THCA) (Figure 1b). Additionally, the results obtained from the Oncomine and GEPIA2 databases were also cross-checked from the GENT2 database. It also provided the SYK gene expression profile in a boxplot of 72 different paired normal vs. cancer tissues, confirming that this gene was significantly upregulated in various cancer types, including CRC. Secondly, the mRNA expression of the SYK gene was analyzed in COAD tissues with their corresponding normal samples in light of different clinicopathological boundaries from the UALCAN and GEPIA2 databases. The results obtained from the UALCAN database indicated that the SYK gene mRNA expression was significantly increased in TCGA COAD samples. The correlation between the *SYK* gene mRNA expression and different clinicopathological parameters in “TCGA” COAD tissues demonstrated that the SYK gene was significantly upregulated based on other variables, such as sample types, individual cancer stages, race, gender, weight, age, histological subtypes, nodal metastasis status, and TP53 mutation status. The data are presented in Figure 2 and Appendix A.

Moreover, the data depicted that the overexpression of the SYK gene was significantly higher in male patients than in female patients and patients between 41–80 years old. This overexpression was significantly higher in African-American patients than in Caucasians and Asians. Likewise, the SYK gene expression in box plots and based on the pathological stages of COAD was also retrieved from the GEPIA2 database. This database also provided similar results to UALCAN for SYK gene expression, and the SYK gene was also overexpressed in COAD samples than in normal tissues in log_2_ scale (*p* < 0.01; Figure 3a).

Similarly, the SYK gene overexpression was also evaluated in the different pathological stages of COAD represented in the violin plot (Figure 3b). The F test value was 0.155, and a probability of the F statistic Pr > F = 0.927. Here, the amount or spread of cancer according to the four cancer stages are depicted in box plots. The wider part in the middle resembles a higher probability of expression around the median value than the skinner part at the end. Apparently, there is no major fluctuation in expression is observed among the four cancer stages. Third, the promoter methylation level of the SYK gene in COAD was determined from the UALCAN database, where the methylation profile is presented on the Beta value scale. The degree of the promoter DNA methylation is shown with a beta value, ranging from 0 (unmethylated) to 1 (completely methylated).

The promoter methylation level of the SYK gene was higher in COAD than in normal tissues based on sample types, individual cancer stages, patient’s race, gender, age, and weight, tumor histology, and T53 mutation status. The data on the promoter methylation level of SYK in COAD is presented in Figure 4 and Appendix A.

### 3.2. Genetic Alteration Analysis in SYK Protein Sequences Associated with Colorectal Cancer Development

Using the cBioPortal information, multiple genetic alterations data were generated to inspect the useful meaning of the SYK gene in CRC development. First, the changes in SYK were queried in this database using 3953 samples of 3806 colorectal malignancy patients from 10 investigations. In 635 amino acid-long human SYK protein, 57 mutations were found, and the somatic mutation frequency in SYK is 1.3% (Table 1).

A lollipop plot depicted these 57 mutations, where 40 were missense type and 17 were truncating type (Figure 5a). Afterward, the genetic alteration recurrence of the SYK gene was analyzed using the information from various colorectal malignancy examinations. From this analysis, it was determined that the SYK alteration frequency fluctuated significantly across different CRC studies. Among these studies, SYK was mostly altered in COAD, with the most notable alteration frequency of 4.17%. On the contrary, the least alteration rate occurred in COAD studies (Figure 5b). 

Lastly, a unique expression analysis was conducted between the SYK mRNA expression and putative CNAs. From this analysis, the level of the SYK mRNA expression was observed, and the amplification was the most upregulated CNA in the RNA Seq V2 RSEM scale. On the other hand, shallow deletion was the most widely recognized kind of alteration (Figure 5c). Overall, it was evident that various genetic alterations were assembled in SYK that led to CRC development. 

### 3.3. Prognostic Value of SYK and Survival Analysis

To further assess the expression of the SYK gene and the clinical prognosis of patients with CRC, several online tools were used, including the GEPIA, OncoLnc, UALCAN, and GENT2 databases. The GEPIA server provides the survival plots of both OS and DFS by examining 135 TCGA COAD tumor samples.

These survival plots were made using a 50% median cut-off value, 95% confidence interval, and Hazard Ratio from GEPIA. The database depicted that a higher OS and DFS were observed in a lower SYK expression level (Figure 6a–d). To make a Kaplan–Meier plot by analyzing a wide range of TCGA cancer studies from the OncoLnc database, both upper percentile and lower percentile were selected as 25. Then, the OS data were analyzed using the CRC patients’ survival data compared to the gene expression data from the UALCAN database. OncoLnc analyzed 110 cancer samples with both low and high SYK expression.

On the contrary, UALCAN analyzed 71 COAD samples with higher expression and 208 samples with lower expression. Based on the graph from the OncoLnc and UALCAN databases, a favorable prognosis was observed in low SYK expression (Figure 6e,f). The Kaplan–Meier plots were provided by the GENT2 database using subtypes and median cut-off, where the colon tissues had the molecular subtype, AJCC stage, Duke stage, grade, and histology categories for subtype analysis. Similar survival was detected for both lower and higher levels of SYK expression (Appendix A). Overall, a positive correlation was noticed between the overexpression of the SYK gene and the poor prognostic predictor for colon cancer. A lower SYK expression level was related to a better prognosis. Therefore, by analyzing the prognostic value and survival plot, it can be concluded that SYK can be considered as a tumor suppressor gene for CRC.

### 3.4. Analysis of Correlated Genes and Preparation of an Interaction Network

For the identification of genes correlated to SYK for COAD, two web-utilities, UALCAN and GEPIA, were used. This correlation was considered to be significant when the Pearson-correlation coefficient value is >0.47 (Appendix A). Both databases showed that SECISBP2 had the highest positive correlation with SYK in COAD. Twenty correlated genes were extracted from both websites to prepare an interaction network. A GeneMANIA web tool was used to investigate the interaction network. The 20 correlated similar genes (TCF7, CDC14B, FKTN, TBC1D13, GLE1, CCDC170, NAA35, TSTD2, GOLGA1, TMEM8B, SNX30, RNF20, EFCAB14, PCDH19, FGGY, SECISBP2, ZNF782, SMC5, C6orf97, and KIAA0494), along with SYK, were employed to build a network (Appendix A). The network was built by an automatically selected weighting method where it showed 79.13% co-expression and 20.87% co-localization in a PPI network.

In addition, the prediction of the PPI network of the SYK gene to express its relationship with significantly correlated genes, the GeneMania web tool was also used. An automatically selected weighting method was selected for the network of 20 genes (HCLS1, LYN, FCGR1A, WIPF1, POU2AF1, GP6, PIK3CG, ERBB4, CD3E, CBL, PIK3CB, EPOR, RPS6KA2, CTTN, NFAM1, BLK, ITGA2B, APBB1IP, FYN, and CD72) related with SYK through several attributes, including co-expression, co-localization, pathway, physical interactions, shared protein domains, predicted, and genetic interactions (Appendix A). This network showed 67.64% of physical interactions, 13.50% of co-expression, 6.35% of predicted, 6.17% of co-localization, 4.35% of the pathway, 1.40% of genetic interactions, and 0.59% of shared protein domains.

### 3.5. Pharmacokinetics Analysis

In this study, six ligands (among them, one was used as the control) were selected based on their ADMET results, and the phytochemicals had a better absorption rate on a human digestive system. In addition, the compounds exhibited a better excretion rate from the body after the distribution and metabolism of ligands; they also had no higher toxic effects on organs (mainly hepatocytes) (Table 2).

### 3.6. Molecular Docking and Post-Docking Data Analysis

The best ligand candidates among the selected aromatic phytochemicals with SYK protein, which bear the PDB ID at 4XG3, were determined using a molecular docking process through the PyRx 0.8 package platform. The PyRx 0.8 tools displayed the best docking score between the macromolecules and ligands. Here, CAP was considered as the control ligand with a docking score of −6.5 Kcal/mol and grid center at X = 36.7526, Y = 3.6328, and Z = 25.3921, along with the grid box dimension of X × Y × Z at 35.9128 Å, 33.0428 Å, and 25.0000 Å, respectively. The aromatic ligand glabridin displayed the best fitting score of −8.2 Kcal/mol using the control drug grid parameters. Furthermore, other potential phytochemicals, such as curcumin, kaempferol, quercetin, and genistein, possessed the best docking score of −8.0, −7.3, −7.2, and −7.1 Kcal/mol, respectively, which are shown in Table 3.

Ligplot+ Version 2.2 showed all the fitting status of the docked complexes, where non-covalent and hydrophobic interactions were determined. The “SYK domain-CAP (control drug)” complex was stabilized by four hydrogen bonds (with Asp612-2.89 Å, Glu614-2.85 Å, Glu614-3.08 Å, and Tyr546-2.79 Å) and eight hydrophobic bond interactions (with Glu452, Glu586, Glu542, Leu453, Lys548, Tyr611, Val613, and Val503). Table 3 also shows that the “SYK domain-glabridin” complex was stabilized by three hydrogen bonds (Asn545-2.94 Å, Tyr611-2.79 Å, and Val613-3.08 Å) and six hydrophobic bonds (Asp612, Gln462, Glu542, Leu453, Pro541, and Tyr546), whereas two hydrogen bonds (Asn524-3.10 Å and Asn524-3.14 Å) and 13 hydrophobic bonds (Asn545, Asp612, Glu542, Glu586, Glu614, Leu453, Lys548, Ser550, Thr504, Tyr525, Tyr546, Tyr611, and Val613) were shown by curcumin against the receptor. Kaempferol formed four hydrogen bonds (Asn524-3.11 Å, Ser550-3.05 Å, and Tyr546-3.18 Å) and six hydrophobic bonds (Glu614, Glu542, Lys548, Phe549, Tyr525, and Val613).

Conversely, quercetin was stabilized by four hydrogen bonds (Asn524-3.06 Å, Asn524-3.16 Å, Asp612-3.06 Å, and Ser550-3.11 Å) and seven hydrophobic bonds (Glu542, Glu614, Lys548, Phe549, Tyr525, Tyr546, and Val613). Finally, genistein formed two hydrogen bonds (Glu614-2.86 Å and Tyr546-3.04 Å) and seven hydrophobic bonds (Asn524, Glu542, Lys548, Ser550, Tyr525, and Val613) with the SYK (PDB ID: 4XG3) receptor molecule. Most importantly, all the “protein-ligand complex” interactions were represented in Figure 7.

### 3.7. Molecular Dynamics Simulation (MDS)

The MDS trajectories were used to explore the multiple simulation descriptors and understand the complexes’ flexibility. Figure 8 shows that the RMSD of the simulation complexes initially experienced an upper trend until 5 ns and stabilized until 30 ns. Therefore, the complexes increased their RMSD profile after 40 ns, but all complexes had a stable profile. The RMSD did not fluctuate much in complexes, except in curcumin and quercetin. Therefore, the other four complexes had a steady trend in RMSD, which indicated the rigidity of the complexes. The complexes had an RMSD < 2.5Å, except for quercetin and curcumin. The RMSF of the amino acid residues from the protein-ligand complexes were checked to understand the flexibility of the amino acid residues. The maximum amino acid residues from the complexes had an RMSF < 2.5Å, which had lower flexibility (Figure 8).

The SASA from the simulation trajectories was analyzed to understand the change in the protein surface area. The kaempferol complex had a higher SASA trend at the initial phase, which indicates the protein surface area expansion. After 30 ns, the kaempferol complex had a similar SASA profile. The glabridin complex had a comparatively lower SASA profile compared to other complexes, and this trend indicates the truncated nature of the complexes, the other complexes had a steady profile and SASA (Figure 8). 

Moreover, the Rg of the protein complexes was checked to understand the protein mobility and rigidness. The protein-ligand complexes had a straight line in the Rg profile, and little fluctuations were observed. The two complexes, quercetin, and kaempferol had a higher Rg trend after 60 ns, which might occur due to the labile nature of these complexes. Therefore, the hydrogen bond of the simulation systems was analyzed as they play an important role in stabilizing the complexes Figure 8. The six complexes had a stable hydrogen bond trend in simulation, and no significant fluctuations were observed.

## 4. Discussion

The mRNA expression patterns of the SYK gene across different types of cancer samples, particularly in CRC samples, with their analogous normal control, were upregulated in BRCA, CESC, CHOL, ESCA, GBM, HNSC, LAML, READ, STAD, and UCEC, and downregulated in BLCA, KIRC, KIRP, PRAD, SKCM, THCA, etc. (Figure 1). Moreover, SYK was significantly upregulated in COAD compared to the normal samples based on different clinicopathological parameters (Figure 2 and Figure 3). Thus, there is a positive correlation between SYK expression and increased risk of tumor metastasis formation. Several reports also demonstrated the role of the SYK gene as a potential prognostic marker in various cancer types, such as human breast carcinomas, cervical carcinogenesis [55], human lung cancer [56], ovarian cancer, and neck cancer, glioblastoma, etc. It is recommended that the abnormal expression level of the SYK gene is associated with CRC development. Recently, SYK has been prominent and may be an effective drug target due to its intimate connection with cell cycle regulation and association with tumorigenesis. Therefore, integrative bioinformatics analysis was performed in this study through several powerful publicly available datasets, indicating that SYK can be a potential prognostic biomarker for CRC disease. 

Here, the divergent role of SYK in cancer might be due to the phosphorylation or methylation at the promoter region. To examine this, we explored the SYK gene promoter methylation level across COAD based on different clinicopathological parameters (Figure 4). The promoter methylation level fluctuates significantly across the various stages of COAD, and the methylation level was higher in tumor samples than in normal samples. Yang et al. (2013) [10] reported a correlation between the clinical relevance and SYK methylation in CRC patients, and they found SYK methylation in 48.6% of CRC tissue samples and 57.1% of cell lines. SYK expression could be resorted by a demethylation agent. However, this required further in-depth investigations.

Furthermore, to investigate the functional relevance of the SYK gene in CRC development, the mutations and CNAs of the SYK protein sequence were analyzed based on ten cancer studies. In total, 57 mutations were found, of which 40 were missense type, and 17 were truncating type (Figure 5). Some researchers also reported that identifying the genomic regions that undergo intermittent alteration might provide a powerful way to discover the oncogenes in human cancers [57]. From this result, it was evident that SYK might play an active role in CRC development. The prognostic relevance of the SYK gene for CRC using a KM plotter from various websites, including GEPIA, OncoLnc, UALCAN, and GENT2, and the analysis of survival curves showed that the lower expression of SYK was related to a higher OS and DFS (Figure 6).

Furthermore, low SYK expression was observed to have a better prognosis based on the data from different colon tissue subtypes, including the molecular subtype, AJCC Stage, Duke stage, grade, and histological subtype (Appendix A). It is necessary to identify the particular gene responsible for the altered gene expression and survival probability differences because that gene could be a cancer biomarker [58]. The validation of potential biomarkers is associated with the survival rate [59]. The SYK gene is a potential marker and tumor suppressor for the reduced expression of human BC and pancreatic ductal adenocarcinoma (PDAC) [8,60]. A positive correlation was also noticed between the overexpression of the SYK gene and the poor prognostic probability for CRC. Besides, the survival plots from various databases suggest that SYK expression can contribute to the development and prognosis of CRC. It can also be considered a tumor suppressor gene. To determine the activity of the SYK gene, the correlation and co-expression were measured using the Pearson-CC. Here, the UALCAN and GEPIA databases were searched to identify the positively correlated genes of SYK in COAD tissues. Twenty correlated genes from both websites were collected to prepare an interaction network from GeneMANIA, where Pearson CC >0.47 was considered as moderately significant. Both databases confirmed that SECISBP2 shared the highest Pearson CC value with SYK. SECISBP2 was considered a novel therapeutic target for DLBCL, and mutations in this gene have been related to an abnormal thyroid hormone metabolism. The interaction showed 79.13% of co-expression, which means that those genes are linked with each other by their expression level in similar conditions, and 20.87% of co-localization (Appendix A), which indicates that the linked genes are expressed in the same tissue or cellular location. To determine another interaction network, GeneMANIA was also employed, and the biologically processed based “automatically selected weighting method” was utilized for the network (Appendix A). All the interacted twenty genes were directly or indirectly connected with different cancer types. HCLS1 contains the potential prognostic characteristics of CRC [61]. Lyn can be used as a primary target for the treatment of hormone-refractory human PC. It is also involved in the activation of CRC [62,63] PTEN was identified as a tumor suppressor and therapeutic target for cancer [64] WIPF1 is a candidate prognostic marker for BC, brain cancer, and CRC [65] ErbB4 gene plays a significant role in BC prognostics and therapy [66]. CBL-b can regulate cancer metastasis [67]. EPOR contributes to BC progression [68]. CTTN can be a new molecular therapeutic target for esophageal squamous cell carcinoma, and it also promotes the proliferation of CRC cells [69,70]. Here, these two categories of protein-protein interaction networks depicted that SYK interacted with these genes that are somehow involved in tumor suppression or tumor promotion.

In recent times, in silico drug design method has become very popular because it increases the speed of drug development by analyzing the result of pharmacophore profiling, molecular docking, post-docking screening, MDS, and prediction of noble compounds against various diseases [71]. Here, five out of six selected phytochemicals were functionally active, and the remaining one phytochemical was used as a control. ADMET profiling is an effective strategy to substantially save the costs of drug development and offer “fact tests” and secondary complimentary opinions for high-performance assays [72,73]. Using pkCSM pharmacokinetics & Swiss ADME online servers, the pharmacokinetic properties of the six selected phytochemicals are shown in Table 2 [74]. Under the pharmacokinetics study, the pharmacophore properties (Lipinski’s rule of five) of all selected ligands were carried out to determine their molecular weight, number of rotatable bonds, log p-value, donors and acceptors of hydrogen bonds, and degree of infringement (violation level). All selected phytochemicals were not permeable to the blood-brain barrier and had better human intestinal absorption, no carcinogenicity, no hepatotoxicity, and no AMES toxicity for both humans and mice, except for the control ligand. After their digestion, a small level of toxic waste was excreted. 

Molecular docking is the process that estimates how two or more molecules bind together with the best structural confirmation and the lowest binding energy. The more significant and efficient drug candidates were selected based on the scoring value of molecular docking using the PyRx virtual screening tool [Version: 0.8] [44,45]. The docking operation for the five phytochemicals and control molecules with receptor SYK kinase found that the docking score of the control was −6.5 Kcal/mol. Among the five phytochemicals, glabridin showed the best binding energy of −8.2 Kcal/mol, whereas genistein was −7.1 Kcal/mol, Curcumin has −8.0 Kcal/mol, and all the docking results were represented via Table 3. Afterward, Ligplot+ [Version: 2.2], an effective investigative tool that generally runs on the Java interface, was used to analyze the 2D protein-ligand interaction scheme. It is important to mention that Ligplot+ only runs the PDB file retrieved from the PyMoL visualizer tool [Version: 2.4.1].

MDS is a diverse technique that investigates the biomolecular interactions and the interaction between the protein structure and function for modern drug discovery and the performance data from dynamic trajectory [75,76]. In this study, the dynamic simulator YASARA (Version 11.9.18) retained all the physiological and physiochemical parameters (Temp, 310 K; pH, 7.4) and compressors (Na+, Cl−) in 100 ns [54], and the performance data assessed the dynamic condition of the selected active compounds to suppress the soluble receptor molecules (SYK). The RMSD, RMSF, Rg, hydrogen bond number, and SASA were also analyzed by this simulation tool [77]. To quantify the protein structure’s stability and predict the conformation shifts, the RMSD [78] of the selected SYK soluble protein backbone was used; the lower value shows the most stable complex. An RMSD value of <1.5 Å typically shows a greater accuracy in the docking concerning the effective binding position. In this study, the RMSD values of protein-ligands interacted in an acceptable range, i.e., minimum values <2 Å (the lowest value for CAP is approximately 1.0 Å), and maximum values ≤3 Å, indicating a better docking position and that the enzyme structure was not disrupted (Figure 8). The RMSF measures the mean protein molecule fluctuations from a reference location, and the residue level fluctuations are portrayed in the RMSF plots.

Furthermore, it is essential to differentiate the local changes along the protein chain [79]. The ligands fluctuated from 1.5 to 2.5 Å, while quercetin and curcumin showed a fluctuation rate of not more than 4 Å, which indicates that the rate of flexibility is a significant amount (Figure 8). Rg measures the distance between the center of mass and terminal protein. Therefore, this parameter measures the compactness of the protein molecule and gives us a deeper understanding of the protein’s folding properties [80]. Additionally, a higher Rg value signifies slackpacking, whereas a lower Rg value reveals compact packing [81]. Figure 8 shows all the Rg values, among which glabridin had the best matching capacity (28.4) compared to the control ligand (CAP, 28.45). The compounds for glabridin and curcumin, alongside their enzymes, have better compactness, and the numerical values are presented in both cases (28.45). The weaker binding capacity was observed in quercetin and genistein (28.39). Similarly, the value of SASA determines the effectiveness of interaction among the macromolecule-ligand complex.

In contrast, the interaction between enzyme surface area and water depicts the amount of energy within per area of ligand and macromolecule [82]. One of the highest SASA values complies with the unstable structure in which hydrophobic amino acid residues are in close contact with the water molecule [83]. According to SASA result, the kaempferol has been followed by the highest values from SASA (26,250 Å2), but CAP, kaempferol, and curcumin have higher SASA values. However, both glabridine and genistein have the lowest SASA values, and they bind precisely to the receptor (Figure 8). The conformational stability of macromolecules and ligands was identified by the complete number of intermolecular hydrogen bonds [84], and the lowest number of hydrogen bonds were observed in kaempferol and glabridin, with a conformational stability of linked macromolecules of 1049 and 928, respectively, which are more stable than that of the control (Figure 8).

In this current in-silico study, there was no sufficient clinical research evidence on CRC by targeting SYK kinase. This study focused on a new insight for CRC treatment in which more wet lab and clinical-based studies would be required to validate these drug-like molecules by targeting SYK kinase protein. As a consequence, these potential bioactive phytochemicals can be used as an alternative therapeutic option for colorectal cancer treatment after the validation of their anticancer activity in vitro and in vivo research models.

## 5. Conclusions

In conclusion, multiple online bioinformatics platforms and web resources were used to systematically analyze the SYK gene expressions, methylations, mutations, CNAs, associated genes, survival status, and prognostic values in various human cancers, especially in CRC. Additionally, the current findings demonstrated the critical role of SYK expression and potential SYK-related pathways in the development of human CRC. This may shed new light on SYK as a novel biomarker and therapeutic target for CRC, thereby assisting in translating genomic information into clinical practice. Furthermore, the ADMET analysis, molecular docking, and molecular dynamic simulation effectively determined the right phytochemicals against targeting the SYK kinase. The ADMET analysis showed their effectiveness as drug molecules with no cellular toxicity. The compounds have the best binding affinity and strong interactions with all or at least one of the catalytic residues of SYK kinase. These protein-ligand complexes also showed several non-covalent interactions, such as hydrogen binding, hydrophobic interactions, and electrostatic interactions. The MDS data findings suggest that the physiological environment had the most stable protein-ligand interactions and interact more frequently with SYK protein via hydrogen bonds. It can be established that most of the identified bioactive compounds would have significant efficacy and could be used as a potential alternative option for SYK kinase target of human colorectal cancer (CRC) treatment after their in vitro and in vivo validation.

## Figures and Tables

**Figure 1 jpm-11-00888-f001:**
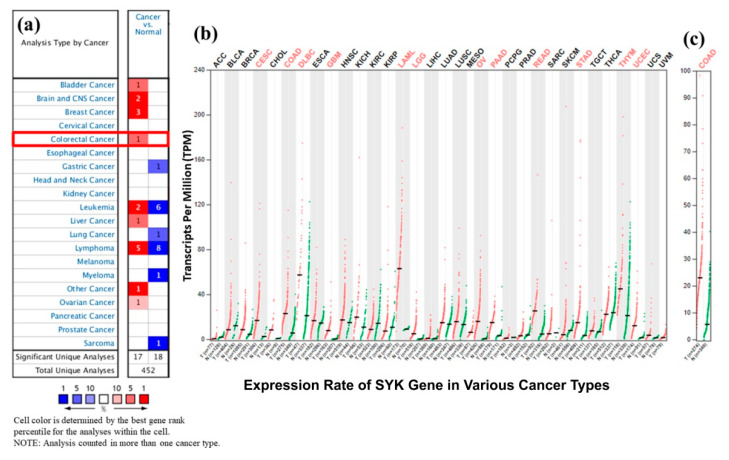
The mRNA expression profile of the SYK gene across various cancer types. (**a**) SYK mRNA expression in different cancer types from the Oncomine database, where the numbered red cells indicate the dataset with significant mRNA overexpression and blue cells indicate mRNA downregulation. (**b**) Pan-cancer expression profile analysis of the SYK gene from the GEPIA2 database in a dot plot, where each dots represent the expression of samples. mRNA expression is presented in transcript per million across different cancer types. (**c**) The transcription level in colorectal adenocarcinoma (COAD) patients with their corresponding normal samples from the GEPIA2 database. The red and green color dots represent tumor samples and paired normal tissues, respectively.

**Figure 2 jpm-11-00888-f002:**
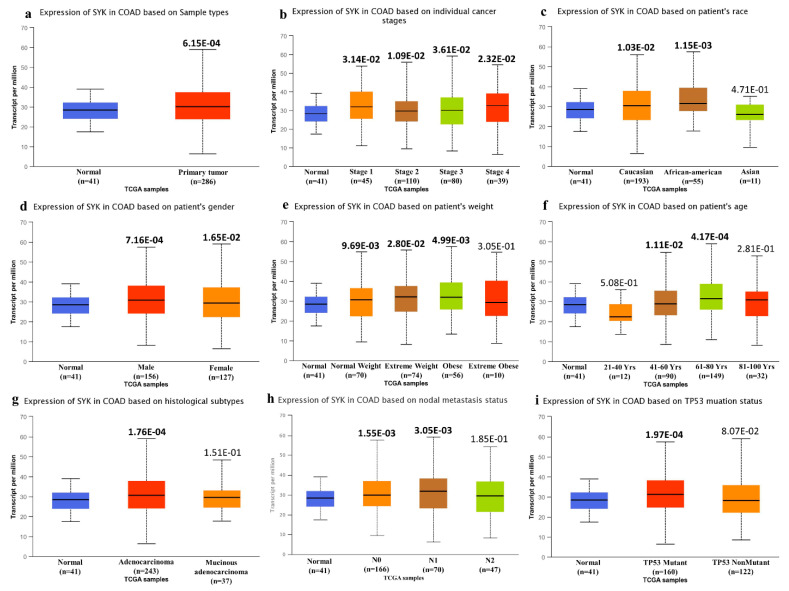
Expression profile analysis of SYK gene in TCGA COAD between normal and different cancer stages based on other variables using the UALCAN database. (**a**) SYK expression in COAD is based on sample types. (**b**) SYK expression in COAD is based on individual cancer stages. (**c**) SYK expression in COAD is based on the patient’s race. (**d**) SYK expression in COAD dependent on the patient’s gender. (**e**) SYK expression in COAD is based on the patient’s weight. (**f**) SYK expression in COAD is based on the patient’s age. (**g**) SYK expression in COAD is based on the histological subtype. (**h**) SYK expression in COAD is based on the nodal metastasis status. (**i**) SYK expression in COAD is based on the TP53 mutation status. Statistically significant *p*-values are marked in bold.

**Figure 3 jpm-11-00888-f003:**
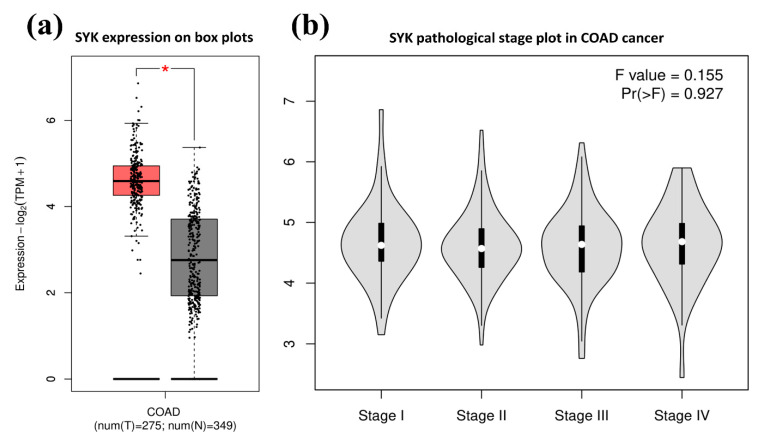
Tissue-wise expression profile of the SYK gene in COAD from the GEPIA2 database. (**a**) The SYK gene expression in COAD on box plots, where the red box indicates the tumor samples, and the black box indicates the normal samples. (**b**) SYK expression in the four tumor stages of COAD on a pathological stage plot. The red star denotes statistical significance, *p*-value < 0.05.

**Figure 4 jpm-11-00888-f004:**
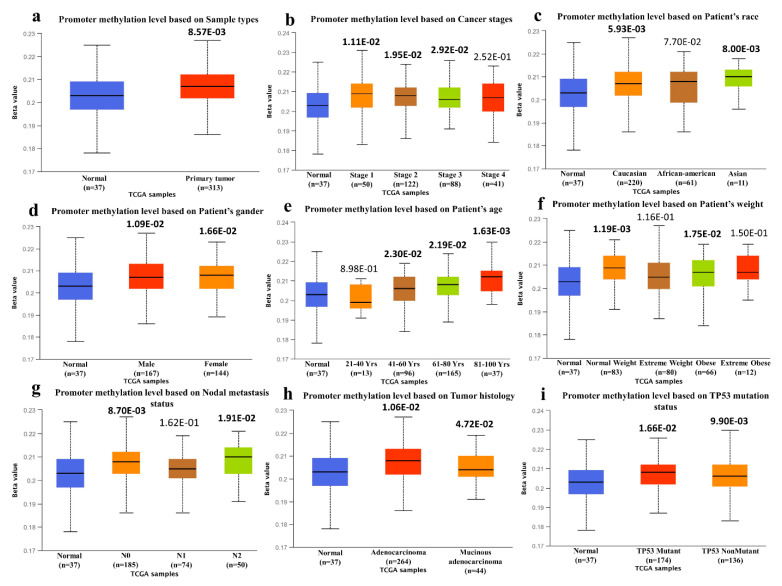
SYK gene promoter methylation level in TCGA COAD based on different variables by using the UALCAN database. (**a**) The promoter methylation level of SYK in COAD is based on sample types. (**b**) The promoter methylation level of SYK in COAD is based on individual cancer stages. (**c**) The promoter methylation level of SYK in COAD is based on the patient’s race. (**d**) The promoter methylation level of SYK in COAD is based on the patient’s gender. (**e**) The promoter methylation level of SYK in COAD is based on the patient’s age. (**f**) The promoter methylation level of SYK in COAD is based on the patient’s weight. (**g**) The promoter methylation level of SYK in COAD is based on the nodal metastasis status. (**h**) Promoter methylation level of SYK in COAD is based on the tumor histology. (**i**) Promoter methylation level of SYK in COAD based on the TP53 mutation status. Based on UALCAN, the beta value indicates the level of DNA methylation, ranging from 0 (unmethylated) to 1 (fully methylated). The bold *p*-value denotes the statistical significance.

**Figure 5 jpm-11-00888-f005:**
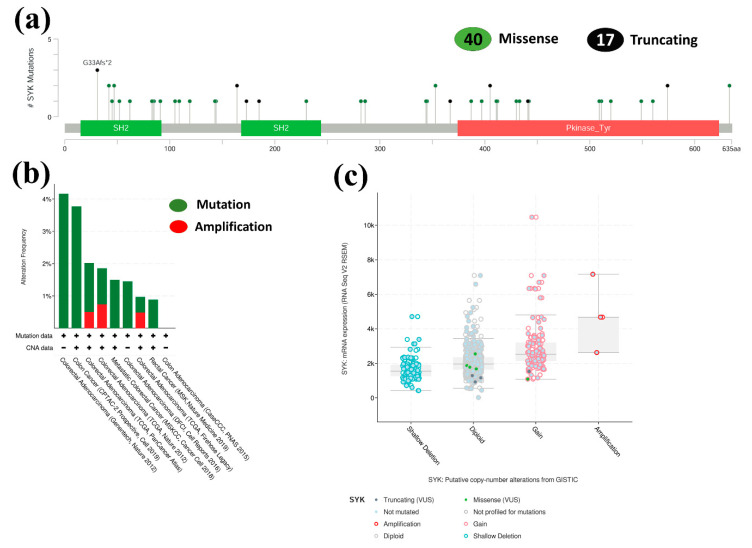
The graphical portrayal of the genetic alteration in the SYK protein sequences based on CRC growth improvement. (**a**) In total, 57 mutations were pointed out in the SYK protein sequence using a lollipop plot. (**b**) Two sorts of alteration frequencies of SYK across different CRC studies were introduced in a bar outline. (**c**) The mRNA expression level of SYK in regard to the different classifications of genetic alterations was addressed in graphical plots dependent on the RNA seq V2 RSEM scale.

**Figure 6 jpm-11-00888-f006:**
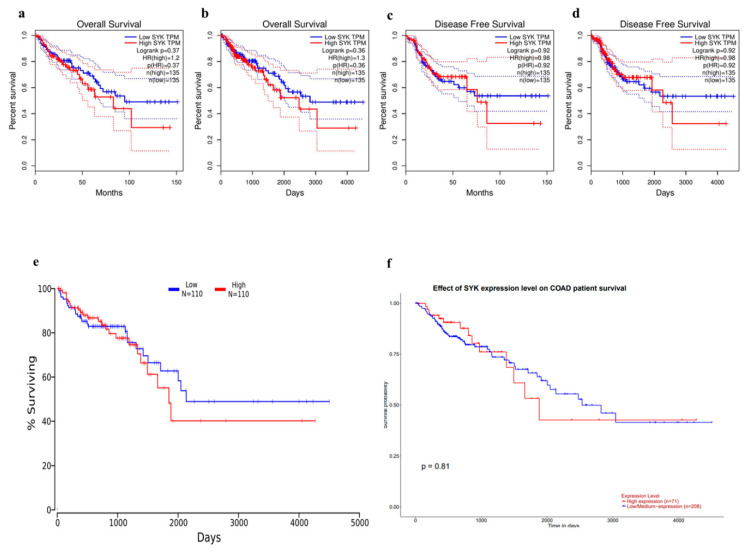
Survival assay for the SYK gene in CRC. (**a**) Survival plot from the GEPIA database. (**b**) Overall survival. (**c**,**d**) Disease-free survival. (**e**) Survival plot from the OncoLnc and (**f**) UALCAN database.

**Figure 7 jpm-11-00888-f007:**
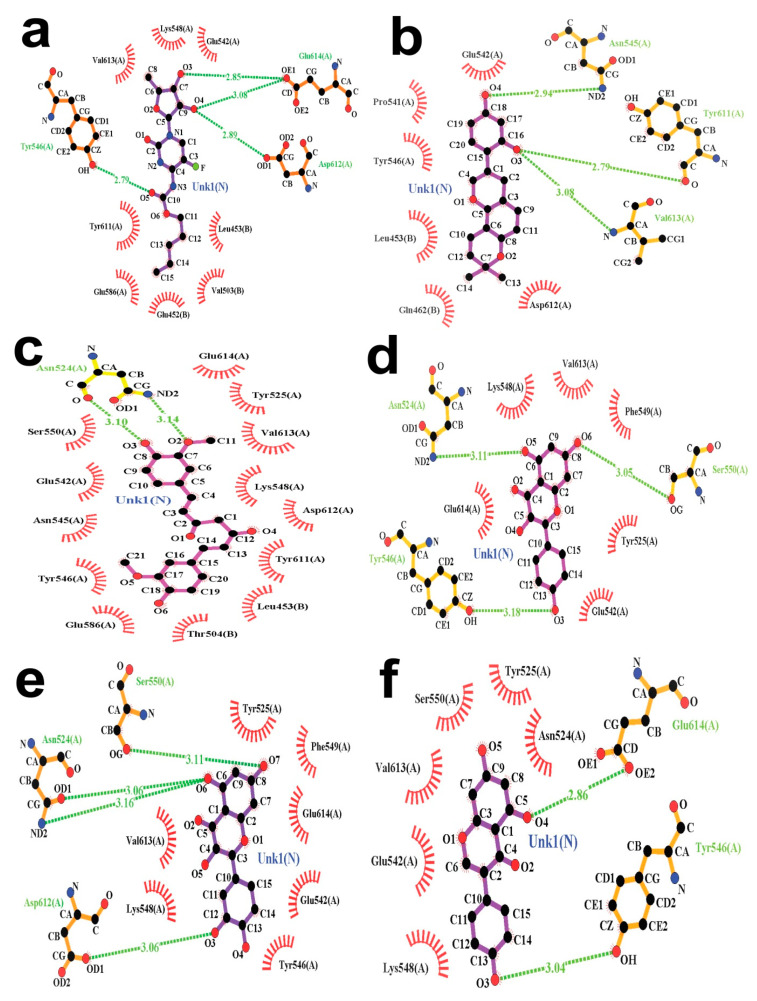
The docked conformation of the “SYK kinase-lead molecule complexes” represents the potential hydrogen and hydrophobic interactions, with hydrogen bonds shown as olive green dotted lines with a specific distance (Å) and hydrophobic interactions shown as red oval-shaped structures. The hydrophobic interactions are shown by thin red lines with ellipses in protein residues. The protein residues that are primarily similar in 3D positions are marked by red spiked arc ellipses. CAP (control)–SYK complex (**a**), Glabridin–SYK complex (**b**), Curcumin–SYK complex (**c**), Kaempferol–SYK complex (**d**), Quercetin–SYK complex (**e**), and Genistein–SYK complex (**f**).

**Figure 8 jpm-11-00888-f008:**
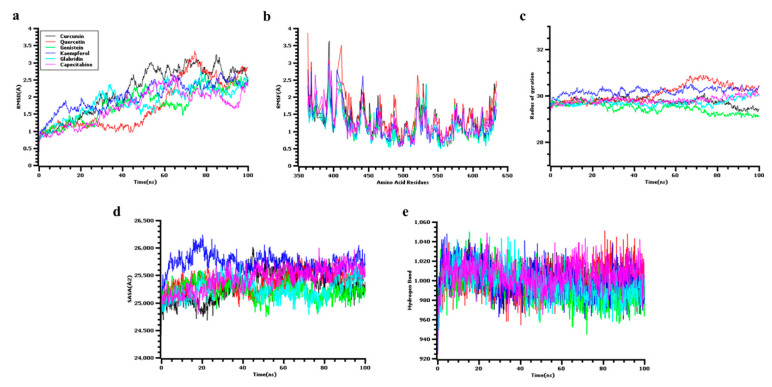
(**a**) The RMSD values were obtained individually by a molecular dynamic simulation of each ligand with the SYK targeted receptor. (**b**) RMSF values were obtained from a molecular dynamic simulation of each ligand when simulated precisely and independently with the target receptor. (**c**) Rg throughout the simulation. (**d**) SASA. (**e**) Maximum number of H-bonds during the simulation.

**Table 1 jpm-11-00888-t001:** A rundown of the hereditary shifts in SYK protein arrangements related to CRC advancement. In total, 57 mutations are detected, including seven duplicate mutations in patients with multiple samples.

Cancer Study	Sample Size	Protein Change	Mutation Type	Sample ID
Colorectal Adenocarcinoma (Genentech, Nature 2012)	74	A52T	Missense	587376
		V633M	Missense	587342
		P85L	Missense	587332
Colorectal Adenocarcinoma (DFCI, Cell Reports 2016)	619	A353V	Missense	coadread_dfci_2016_68
		S511N	Missense	coadread_dfci_2016_1762
		L143M	Missense	coadread_dfci_2016_197
		N406Tfs*13	FS del	coadread_dfci_2016_207430
		N406Tfs*13	FS del	coadread_dfci_2016_2944
		R520H	Missense	coadread_dfci_2016_2372
		P430L	Missense	coadread_dfci_2016_3048
		K397E	Missense	coadread_dfci_2016_3670
		R109Q	Missense	coadread_dfci_2016_62
Colorectal Adenocarcinoma (TCGA, Firehose Legacy)	640	M166Nfs*14	FS ins	TCGA-AG-A02N-01
		K387N	Missense	TCGA-AF-3913-01
		R574*	Nonsense	TCGA-AG-A002-01
Colorectal Adenocarcinoma (TCGA, Nature 2012)	276	M166Nfs*14	FS ins	TCGA-AG-A02N-01
		K387N	Missense	TCGA-AF-3913-01
		R574*	Nonsense	TCGA-AG-A002-01
Colorectal Adenocarcinoma (TCGA, PanCancer Atlas)	594	A353T	Missense	TCGA-D5-6922-01
		M166Nfs*14	FS ins	TCGA-AG-A02N-01
		R574*	Nonsense	TCGA-AG-A002-01
		R574*	Nonsense	TCGA-F5-6814-01
		F549L	Missense	TCGA-F5-6814-01
		K105N	Missense	TCGA-AG-A00Y-01
		D344G	Missense	TCGA-F5-6814-01
		P119S	Missense	TCGA-A6-2686-01
		Y91C	Missense	TCGA-AY-6197-01
		T345R	Missense	TCGA-G4-6586-01
		E442Sfs*31	FS del	TCGA-WS-AB45-01
Metastatic Colorectal Cancer (MSKCC, Cancer Cell 2018)	1134	R42C	Missense	P-0004602-T01-IM5
		R42C	Missense	P-0005230-T01-IM5
		R45C	Missense	P-0013492-T01-IM5
		G33Afs*2	FS del	P-0006365-T01-IM5
		G33Afs*2	FS del	P-0013876-T01-IM5
		A286V	Missense	P-0010929-T01-IM5
		E442K	Missense	P-0010587-T01-IM5
		A282V	Missense	P-0005443-T01-IM5
		S84T	Missense	P-0005823-T01-IM5
		E144G	Missense	P-0010581-T01-IM5
		R367*	Nonsense	P-0006960-T01-IM5
		P411L	Missense	P-0006960-T01-IM5
		V560A	Missense	P-0006960-T01-IM5
		H62R	Missense	P-0005455-T01-IM5
		V433M	Missense	P-0005455-T01-IM5
		G185*	Nonsense	P-0002671-T01-IM3
		R175Gfs*4	FS del	P-0002671-T01-IM3
		Y47N	Missense	P-0000769-T01-IM3
		K509R	Missense	P-0001500-T03-IM5
		M166Nfs*14	FS ins	P-0007831-T01-IM5
		E230G	Missense	P-0013227-T01-IM5
Rectal Cancer (MSK, Nature Medicine 2019)	339	V633M	Missense	RC-MSK-008-pt
		V633M	Missense	RC-MSK-008-tm
		K509R	Missense	P-0001500-T03-IM5
Colon Cancer (CPTAC-2 Prospective, Cell 2019)	110	G33Afs*2	FS del	05CO041
		A83T	Missense	01CO014
		A412S	Missense	05CO015
		Y47C	Missense	11CO059

**Table 2 jpm-11-00888-t002:** The complete pharmacophore and pharmacokinetic profiling of the selected ligands.

Name of Phytochemicals/ADMET Values	Physiochemical Properties					Pharmacokinetics Properties							
	MO	HBA	HBD	Log P	RB	IAb	TCl	LD50	HPT	AMT	MTHD	CaP	CTOR
Capecitabine (Control)	359.354	8	3	0.7602	6	68.027	1.054	2.459	Yes	No	1.051	0.255	2.401
Glabridin	324.376	4	2	4.0007	1	94.164	0.121	2.523	No	No	−0.395	1.284	1.148
Curcumin	368.385	6	2	3.3699	8	82.19	−0.002	1.833	No	No	0.081	−0.093	2.228
Kaempferol	286.23	6	4	2.2824	1	74.29	0.477	2.449	No	No	0.531	0.032	2.505
Quercetin	302.238	7	5	1.988	1	77.207	0.407	2.471	No	No	0.499	−0.229	2.612
Genistein	270.24	5	3	2.5768	1	93.387	0.151	2.268	No	No	0.478	0.9	2.189

Units of Physiochemical Properties: MO, molecular weight (g/mol); HBA, hydrogen bond acceptor; HBD, hydrogen bond donor; LogP, estimated octanol/coefficient of liquid fraction; RB, Rotatable bonds. Units of Pharmacokinetics Properties: IAb, intestinal absorption (% absorbed); TCl, total clearance (log mL/min/kg); LD50, acute toxicity of the oral rat; HPT, hepatotoxicity; AMT, AMES toxicity; MTHD, Maximum tolerated human dose (log mg/kg/day); CaP, Caco2 permeability (log Papp in 10^−6^ cm/s); CTOR, chronic toxicity in oral rat (log mg/kg_bw/day).

**Table 3 jpm-11-00888-t003:** Binding affinity of ligands with SYK (PDB ID: 4XG3) receptor macromolecule and the comprehensive intermolecular interaction between them as compared with the control molecule.

Ligands Name	Binding Affinity (Kcal/mol)	Amino Acid Involved Interaction	
		Hydrogen Bond Interaction	Hydrophobic Bond Interaction
Capecitabine (Control)	−6.5	Asp612 (2.89 Å), Glu614 (2.85 Å), Glu614 (3.08 Å) and Tyr546 (2.79 Å)	Glu452, Glu586, Glu542, Leu453, Lys548, Tyr611, Val613, Val503
Glabridin	−8.2	Asn545 (2.94 Å), Tyr611 (2.79 Å) and Val613 (3.08 Å)	Asp612, Gln462, Glu542, Leu453, Pro541, Tyr546
Curcumin	−8.0	Asn524 (3.10 Å) andAsn524 (3.14 Å)	Asn545, Asp612, Glu542, Glu586, Glu614, Leu453, Lys548, Ser550, Thr504, Tyr525, Tyr546, Tyr611, Val613
Kaempferol	−7.3	Asn524 (3.11 Å), Ser550 (3.05 Å) and Tyr546 (3.18 Å)	Glu614, Glu542, Lys548, Phe549, Tyr525, Val613
Quercetin	−7.2	Asn524 (3.06 Å), Asn524 (3.16 Å), Asp612 (3.06 Å), and Ser550 (3.11 Å)	Glu542, Glu614, Lys548, Phe549, Tyr525, Tyr546, Val613
Genistein	−7.1	Glu614 (2.86 Å) and Tyr546 (3.04 Å)	Asn524, Glu542, Lys548, Ser550, Tyr525, Val613.

## Data Availability

The datasets used and/or analyzed during this study are available from the corresponding authors upon request.

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
