# Peer review of "Analysis of SYK Gene as a Prognostic Biomarker and Suggested Potential Bioactive Phytochemicals as an Alternative Therapeutic Option for Colorectal Cancer: An In-Silico Pharmaco-Informatics Investigation"

_jpm, 2021, doi:10.3390/jpm11090888_

Round 1
Reviewer 1 Report
In paragraph 2.8, please include the PDB IDs of the docked proteins.
The x-axis of fgigure should be edited.
The quality of figure 7 should be improved.
The style of the references should be edited.
Author Response
In paragraph 2.8, please include the PDB IDs of the docked proteins.
>>Response: In our updated manuscript, we have added the PDB ID of the targeted protein in the page 08, line 235 and 236.
The x-axis of fgigure should be edited.
>>Response: We have added the x-axis information in the Figure 01, page 10.
The quality of figure 7 should be improved.
>>Response: We have improved the quality of figure 07 in our updated manuscript in page 28.
The style of the references should be edited.
>>Response: We have reformed all the references according to the Journal Guideline by using EndNote Software.
Reviewer 2 Report
In this study the authors assess the role of SYK expression in the prognosis of colorectal cancer and explore the putative use of selected phytochemical bioactive compounds for CRC treatment. The study is a systematic bioinformatics analysis and all data presented are in silico findings.
- The abstract needs to be re-written, including brief explanations of the rationale of the study. In addition, some clarifications are necessary (for examples, please refer to the uploaded file).
- The manuscript contains some grammar and syntax errors that need to be corrected and some sentences that need rephrasing (for examples, please refer to the uploaded file).
- In Figure 1 color coding (green-red) needs to be added.
- The findings presented in Figure 3b need to be described in the relevant text of the Results section.
- The Discussion section is entirely too long. It contains a lot of information concerning results, giving the impression that it is even more detailed that the Results section itself. Some parts need to be deleted or move to the Results section.
- Most importantly, my major concern is that functional experiments are totally missing from the study. This is something the authors themselves discuss in ln647-649. I can understand that this is mainly an in silico study, but I would really much appreciate the presence of some experimental assays that would fortify the findings and the conclusions of the study: a. SYK overexpression at the mRNA and protein level should be validated in colorectal cancer cells in comparison to normal cells in vitro; b. validation of protein-protein interactions of SYK with selected proteins revealed by the bioinformatics analysis should be performed exdperimentally, so that a molecular mechanism of SYK role in CRC could be proposed; c. a viability and or proliferation assay (e.g. MTT assay) should be included to test the efficacy of the bioactive compounds in colorectal cancer cells in comparison to normal cells; the effect of the investigated bioactive compounds on selected proteins that were fould to interact with SYK could be explored.

Author Response
In this study the authors assess the role of SYK expression in the prognosis of colorectal cancer and explore the putative use of selected phytochemical bioactive compounds for CRC treatment. The study is a systematic bioinformatics analysis and all data presented are in silico findings.
01) The abstract needs to be re-written, including brief explanations of the rationale of the study. In addition, some clarifications are necessary (for examples, please refer to the uploaded file).
>>Response: We have re-written the abstract according to the reviewer suggestions.
02) The manuscript contains some grammar and syntax errors that need to be corrected and some sentences that need rephrasing (for examples, please refer to the uploaded file).
>>Response: We have corrected all the grammar and syntax errors and also rewritten the rephrasing issues according to the reviewer guideline and marked in blue color.
03) In Figure 1 color coding (green-red) needs to be added.
>>Response: we have added the color coding (green-red) in the updated manuscript (page 10, line 287-288).
04) The findings presented in Figure 3b need to be described in the relevant text of the Results section.
>>Response: we have described the findings presented in Figure 3b in the page 13, line 331-337.
05) The Discussion section is entirely too long. It contains a lot of information concerning results, giving the impression that it is even more detailed that the Results section itself. Some parts need to be deleted or move to the Results section.
>>Response: In our updated manuscript, we have erased the unnecessary portion from the discussion part and at the same time added some necessary portions.
06) Most importantly, my major concern is that functional experiments are totally missing from the study. This is something the authors themselves discuss in ln 647-649. I can understand that this is mainly an in silico study, but I would really much appreciate the presence of some experimental assays that would fortify the findings and the conclusions of the study:
- SYK overexpression at the mRNA and protein level should be validated in colorectal cancer cells in comparison to normal cells in vitro;
- validation of protein-protein interactions of SYK with selected proteins revealed by the bioinformatics analysis should be performed experimentally, so that a molecular mechanism of SYK role in CRC could be proposed;
- a viability and or proliferation assay (e.g. MTT assay) should be included to test the efficacy of the bioactive compounds in colorectal cancer cells in comparison to normal cells; the effect of the investigated bioactive compounds on selected proteins that were found to interact with SYK could be explored.
>>Response: Thank you for your comment and appreciation. Basically, our study was based on in- silico approaches that you recognized in main manuscript. We got your point and let you know that undoubtedly experimental study may enhance the methodological accuracy. However, we could not do it due to some limitations. Hopefully, we will conduct all of these wet lab experiments in our next project and submit to the Journal of Personalized Medicine.
Round 2
Reviewer 2 Report
This is the revised version of a previously submitted manuscript.
The authors have addressed most of my points raised during the previous round of the reviewing process.
This manuscript is a resubmission of an earlier submission. The following is a list of the peer review reports and author responses from that submission.
Round 1
Reviewer 1 Report
Line 30: correct "fetal" in "fatal".
In the Introduction section, please descrbe the phytochemicals studied in the present manuscript.
Regarding the docking analysis, the affinity constants could be reported also as molar concentrations. It could be easier to compare with experimental data. Regarding the figures of docking analysis, the quality seems poor.
In the masnuscript, it is important to include the PDBIDs of the docked proteins.
The reference section has to be formatted according to journal guidelines.
Author Response
First and foremost, we'd like to express our heartfelt appreciation for the reviewer's time and effort in reviewing our manuscript.
- Line 30: correct "fetal" in "fatal".
>> Response: We have checked and corrected it (page 2, line 30).
- In the Introduction section, please describe the phytochemicals studied in the present manuscript
>> Response: We have broadly described about the phytochemical and their background study in the Introduction section (page 5, line 122-137) by reviewing a number of literatures.
- Regarding the docking analysis, the affinity constants could be reported also as molar concentrations. It could be easier to compare with experimental data. Regarding the figures of docking analysis, the quality seems poor.
>> Response: In our Research Study, we have used the PyRx 0.8 package platform for molecular docking and this software gives us data in the Kcal/mol unit by default which have been mentioned in the main manuscript (page 26, line 454, 457, 459; page 33, line 614, 615). We did not get any result of molecular docking as a molar concentration unit by using this software as well as from any other published articles in any journals of this field.
All the figures of this manuscript have been checked and increased the image quality.
- In the masnuscript, it is important to include the PDBIDs of the docked proteins.
>> Response: We have added more information about the PDB ID of the docked protein (page 8, line 235; page 26, line 451; page 27, line 473; page 28, line 480)
- The reference section has to be formatted according to journal guidelines.
>> Response: By Following the Journal Guideline, we have further formatted all the references of this manuscript by using the EndNote X9 (Paid and updated Version) references organizing software.

Reviewer 2 Report
This study provides potentially crucial data about SYK, a controversial tumor-suppressor gene, which paradoxycally, if inhibitied, could suppress tumor growth and progression (just as a oncogene).
The study suffers from a number of issues:
-English language. The whole manuscript should be revised by a language editor. This fact is the major flaw in the study; it hinders the overall quality and accessibility to the reader. I am not going to show, point by point, the questiable usage of the language in the manuscript. The title itself could be improved.
-Overall confusionary layout and absence of discussion about the ambivalence of SYK. The authors write about this gene and its protein product freely as a tumor suppressor gene and an oncogene. I believe that an introduction on the controversy regarding this gene would be essential.
-Figure 7 and 8 should be revised, and, possibly, put in the supplementary materials. Figure 7 is too overcrowded with survival graphs, which are barely readeable. Figure 8 offers scarce insights on the issue; it could be improved by focusing on a selection of interactions of key-genes and pathways related to cancer.
Positive aspects of the manuscript:
-The bioinformatics data are well gathered and analyzed. I understand that trying to shred light on a thorny field like the role of SYK in cancer is troublesome, and I think that the ultimate goal of the authors (to suggest some potential treatments for colorectal cancer regarding SYK inhibition) is important.
Author Response
This study provides potentially crucial data about SYK, a controversial tumor-suppressor gene, which paradoxycally, if inhibitied, could suppress tumor growth and progression (just as a oncogene).
The study suffers from a number of issues:
-English language. The whole manuscript should be revised by a language editor. This fact is the major flaw in the study; it hinders the overall quality and accessibility to the reader. I am not going to show, point by point, the questiable usage of the language in the manuscript. The title itself could be improved.
>>Response: We massively revised and improved the quality of our manuscript by the professional language editors (Company name: enago, Ref#: INQ-6153200121_BONBOK-1 with certificate of prove).
-Overall confusionary layout and absence of discussion about the ambivalence of SYK. The authors write about this gene and its protein product freely as a tumor suppressor gene and an oncogene. I believe that an introduction on the controversy regarding this gene would be essential.
>>Response: We have included the dual role of SYK both as tumor promoter and tumor suppressor in the introduction section (page 4, line 104-110). We have introduced the distinct role of this gene in diverse carcinomas either as a tumor promoter or as a tumor suppressor.
-Figure 7 and 8 should be revised, and, possibly, put in the supplementary materials. Figure 7 is too overcrowded with survival graphs, which are barely readeable. Figure 8 offers scarce insights on the issue; it could be improved by focusing on a selection of interactions of key-genes and pathways related to cancer.
>>Response: we have enhanced the quality of the figures (figure 7 and 8). Figure 7 is no longer overcrowded and it is readable now. Also, we moved the figure 7 and 8 to the supplementary file. We have provided the interaction network of the SYK gene to establish the connection of the SYK gene with the CRC more strongly. By analyzing the interaction network, it is evident that the connection of SYK with the other genes that are directly or indirectly related to various cancer. Besides we identified positively correlated genes of SYK in COAD tissues. That’s why we used this interaction network.
Positive aspects of the manuscript:
-The bioinformatics data are well gathered and analyzed. I understand that trying to shred light on a thorny field like the role of SYK in cancer is troublesome, and I think that the ultimate goal of the authors (to suggest some potential treatments for colorectal cancer regarding SYK inhibition) is important.
>> Response: We are grateful to reviewer wonderful comments regarding our Research Article.

Round 2
Reviewer 2 Report
The manuscript was improved since the last version, and I think that it is now suitable for publication.
Author Response
The manuscript was improved since the last version, and I think that it is now suitable for publication.
>>Response: First and foremost, we'd like to express our heartfelt appreciation for the reviewer’s time and effort in reviewing our manuscript.